# Optimization of Convolutional Neural Networks Architectures Using PSO for Sign Language Recognition

Jonathan Fregoso, Claudia I. Gonzalez * and Gabriela E. Martinez

Division of Graduate Studies and Research, Tijuana Institute of Technology, Tijuana 22414, Mexico; jonathan.fregoso@tectijuana.edu.mx (J.F.); gmartinez@tectijuana.mx (G.E.M.)
* Correspondence: cgonzalez@tectijuana.mx

**Abstract:** This paper presents an approach to design convolutional neural network architectures, using the particle swarm optimization algorithm. The adjustment of the hyper-parameters and finding the optimal network architecture of convolutional neural networks represents an important challenge. Network performance and achieving efficient learning models for a particular problem depends on setting hyper-parameter values and this implies exploring a huge and complex search space. The use of heuristic-based searches supports these types of problems; therefore, the main contribution of this research work is to apply the PSO algorithm to find the optimal parameters of the convolutional neural networks which include the number of convolutional layers, the filter size used in the convolutional process, the number of convolutional filters, and the batch size. This work describes two optimization approaches; the first, the parameters obtained by PSO are kept under the same conditions in each convolutional layer, and the objective function evaluated by PSO is given by the classification rate; in the second, the PSO generates different parameters per layer, and the objective function is composed of the recognition rate in conjunction with the Akaike information criterion, the latter helps to find the best network performance but with the minimum parameters. The optimized architectures are implemented in three study cases of sign language databases, in which are included the Mexican Sign Language alphabet, the American Sign Language MNIST, and the American Sign Language alphabet. According to the results, the proposed methodologies achieved favorable results with a recognition rate higher than 99%, showing competitive results compared to other state-of-the-art approaches.

**Keywords:** PSO; sign language recognition; optimization of convolutional neural networks

## 1. Introduction

Deep neural networks have demonstrated their capacity to solve classification problems using a hierarchical model, millions of parameters, and learning with big databases. Convolutional neural networks (CNN) are a special class of deep neural networks that consist of several convolutions, pooling, and fully connected layers; this has proven to be a robust method for image or video processing, classification, and pattern recognition. In recent years CNN has attracted attention for achieving superior results in various applications in the computer vision domain, such as medicine, aerospace, natural language processing and robotics [1,2].

CNN are widely used in the field of industry, however, when designing CNN architectures, we face some challenges which include the high computational costs for information processing and finding the optimal CNN parameters (architecture) for each problem [3]. CNN architectures are made up of numerous parameters and, depending on their configuration, can generate a variety of classification results when applied to solve the same tasks; the setting of the hyper-parameter values is usually based on a random search, performing several tests or adjusting manually and this represents a complex search process. To solve this challenge, various researchers have proposed the implementation of evolutionary

computation approaches to automatically design the optimal CNN architectures and to increase its performance [4,5]. In Sun et al. [6,7], an evolutionary approach is implemented to automatically obtain CNN architectures, achieving good results against the state-of-the-art architectures. In Ma et al. [8], the authors present an analysis of different methodologies based on evolutionary computing techniques to optimize CNN architectures, these were tested on benchmark data sets and achieved competitive results. Baldominos et al. [9] implement an approach to automatically design CNN architectures, using genetic algorithms (GA) in conjunction with grammatical evolution.

In the state of the art, we can find a variety of meta-heuristics that are applied to optimize CNN hyper-parameters, including the FGSA [10–12], harmonic search (HS) [13], differential evolution (DE) [14], microcanonical optimization algorithm [15], Whale optimization algorithm [16] and tree growth algorithm framework [17] to mention a few.

In other research works, the PSO algorithm is used to optimize CNN architectures, obtaining favorable results in the solution of different applications. In Sun et al. [18], Singh et al. [19] and Wang et al. [20], PSO is applied to automatically design CNN architectures; these approaches are tested on known benchmark datasets, and the results obtained are competitive against the state-of-the-art architectures. Besides this, PSO has been implemented in other fields of machine learning, including the optimization of different types of artificial neural network architectures, given favorable solutions for a plethora of problems [21,22]. In [23], PSO optimizes models of modular neural networks and is applied to obtain the blood pressure trend. In [24], a hybrid ANN-PSO method is applied to model the electricity price forecasting for the Indian energy exchange. As well as in [25], the PSO variants are applied to generate optimal modular neural network architectures obtaining competitive results for human recognition. In [26], the PSO algorithm is used to optimize deep neural network architectures and is tested in image classification tasks. In [27] a new paradigm of hybrid classification based on PSO is presented, which is applied for the prediction of medical diagnoses and prognoses. Furthermore, in [28] the artificial bee colony (ABC) [29] and PSO are used to optimize multilayer perceptron neural networks (MLP); the approach is applied to estimate the heating load and cooling load of energy efficient buildings; and the authors report that PSO outperforms ABC, improving the MLP performance. In the listed works, we can note the advantages that PSO offers in the optimization process, increasing performance in different tasks.

In research related to CNN approaches applied to the recognition of sign language, we find the work presented in [30] where a CNN model with stochastic pooling is implemented in the recognition of the Chinese sign language spelling, achieving a rate of $89.32 \pm 1.07\%$ recognition. In [31] a CNN method for Arabic sign language (ArSL) recognition was applied, where the authors report a value of 90.02% precision. In [32] a 3D-CNN approach is applied to sign language recognition for extensive vocabulary, images are captured through a Kinect, the authors report effectiveness of 88.7%.

The contribution of this research work focuses on implementing a hybrid methodology, where the PSO algorithm is applied to find the optimal design of parameters for CNN architectures. This work presents two optimization approaches; in both, the parameters considered are the number of convolution layers, the filter size used in each convolutional layer, the convolution filters number and, the batch size. In the first approach, the consistency of the parameters between each layer is maintained in the same conditions and the objective function is given by the recognition rate. In the second approach, the aim is to find more random searches in the architectures that the PSO produces; in this case, the values for each convolution layer are completely different, and the objective function is given by the highest recognition rate, and the lowest Akaike information criterion (AIC); the latter helps to obtain more robust performance of the network with the minimum parameters as the AIC allows penalizing the number of parameters used in each training. The optimized architectures are tested with three sign language databases, including the Mexican Sign Language (MSL) alphabet, the American Sign Language (ASL) alphabet [33], and the American Sign Language MNIST (ASL MNIST) [34]. This research aims to impulse

the investigation in the soft computing area for the development of tools to help the deaf community for a more inclusive society [35].

The structure of this work is organized as follows. Section 2 presents the general theory about convolutional neural networks. Section 3 introduces PSO theory, including definitions, functionality, and the main equations. Section 4 details the methodology for developing the two PSO-CNN optimization approaches. Section 5 describes an analysis of the experimental results achieved after the optimized architectures are implemented for the three databases. Additionally, Section 5 presents a statistical test to compare the two optimization proposals, and we also show a comparative analysis against other CNN approaches focused on sign language recognition. Finally, Section 6 gives important conclusions and future works.

## 2. Convolutional Neural Networks

Biologically inspired computational models are capable of far outperforming previous forms of common artificial intelligence of machine learning. One of the most impressive forms of ANN (artificial neural network) architecture is that of CNN, which is mainly implemented to solve difficult image-based pattern recognition tasks.

CNNs are a specialized type of ANN with supervised learning, which process their layers by emulating the visual cortex of the human eye. This procedure allows the recognition of characteristic patterns in the input data, which makes it possible to identify objects through a set of hidden layers, which have a hierarchy and are specialized. The first layers are capable of detecting curves and lines and to the extent that you work with deeper layers, it is possible to achieve the recognition of more complex shapes, such as a silhouette or peoples' faces.

These types of networks are designed to operate specifically with image processing. The design of its architecture emulates the behavior of the visual cortex of the brain when processing and recognizing images [36]. Its main function is to locate and learn the information characteristic patterns, such as curves, lines, color tones, etc., through the application of convolution layers, which facilitate the process of identification and classification of objects [37,38].

The basic CNN architecture is presented in Figure 1, which consists of five layers: the input, convolution, non-linearity (ReLu), pooling, and classification layer [39,40], these are described in the following subsections.

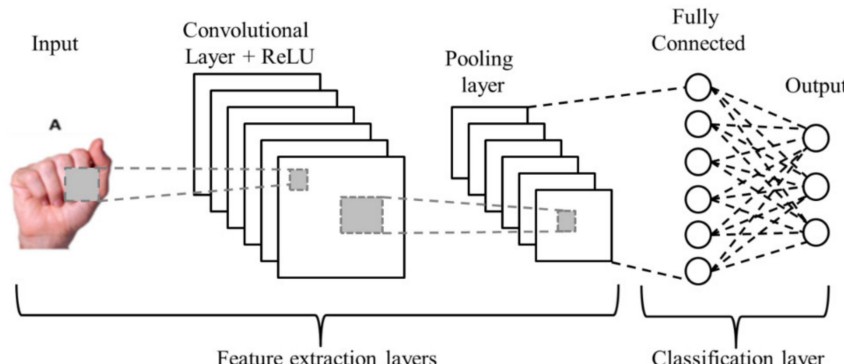

**Figure 1.** The minimal architecture of a CNN.

CNNs are widely implemented in applications that need the use of artificial vision techniques. Although the results that have been obtained are very promising, the reality is that they incur high computational costs; therefore it is essential to implement techniques that allow your performance to be increased. For this reason, an optimization of the CNN parameters is presented to improve the recognition percentage and reduce computational cost. In Figure 2, we can appreciate some parameters that can be optimized in each CNN layer [41].

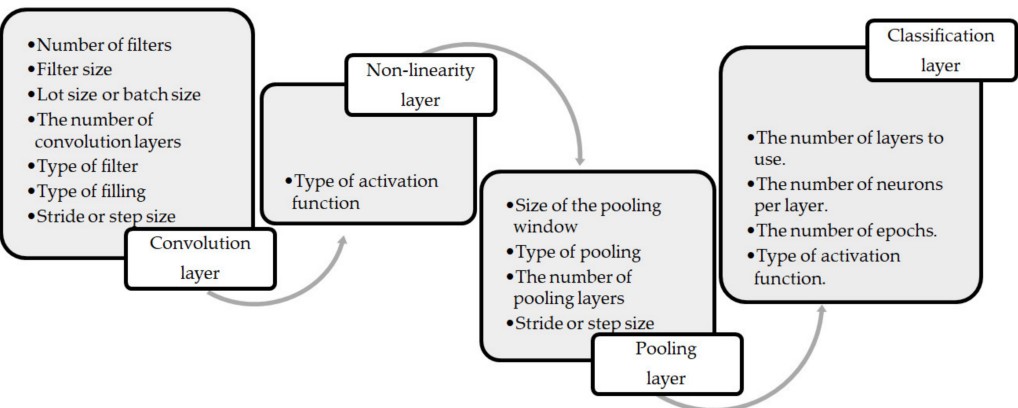

**Figure 2.** Layers and the parameters per layer of a CNN.

### 2.1. Input Layer

It is the first layer of a CNN, here the images or videos are entered that are going to be processed by the neural network to extract their characteristics. All information is stored in two-dimensional matrices. To increase the effectiveness of the algorithms and reduce the computational cost, it is recommended to carry out a previous preprocessing of the images to be trained, such as segmentation, normalization of pixel values, extraction of characteristics of the objects or the background to keep the most relevant information, working them in grayscale, etc.

### 2.2. Convolution Layer

One of the most distinctive processes of this type of network is convolutions. It consists of taking a group of pixels from the input image and making a dot product with a kernel to produce numerous images that are the feature maps; these maps are distinct and depend on the type and size of the convolution filter implemented in the image.

Among the important characteristics that it gives to the kernel, is to detect lines, edges, focus, blur, curves, colors, among others. This is achieved by performing the convolution between the image and the kernel, multiplying the filter values pixel by pixel with those of the image, by traveling the filter from left to right; this representation can be appreciated in Figure 3 [42], where * stands the convolution operation.

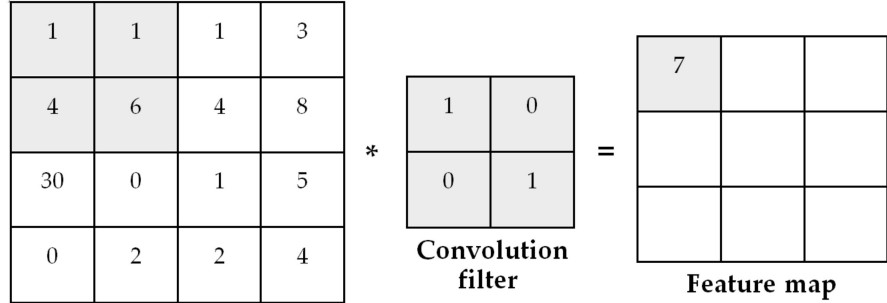

**Figure 3.** Feature maps generated by the convolution process.

### 2.3. Non-Linearity Layer

The activation function in the convolutional layer has the same proposal that the activation used in any neural network, commonly a non-linearity function is used to normalize the images. There exist different activation functions; one of the most used in this type of models is the rectified linear unit (ReLU) function which brings back a value of zero if it receives a value less than zero as input, nevertheless for any value greater than zero the same parameter comes back [41,42].

### 2.4. Pooling Layer

The pooling task is used to reduce the dimensionality of the network, in other words, allows the reduction of the number of parameters, which shortens training time and combats over-fitting [41]. Among the most used types of grouping, we can mention the following: (1) mean, select the arithmetic mean of the values, (2) max pooling, select the pixel with the largest value in the feature map and (3) sum, take the sum of all the elements present in the feature map.

The pooling operation is usually done using a $2 \times 2$ filter, assuming that we have a $4 \times 4$ future map (obtained after the convolution layer), and this operation is carried out; first, the future map is divided into 4 segments with the size of the filer ($2 \times 2$), second, in each segment an pixel value is selected according to pooling type (mean, max, sum). An example is illustrated in Figure 4.

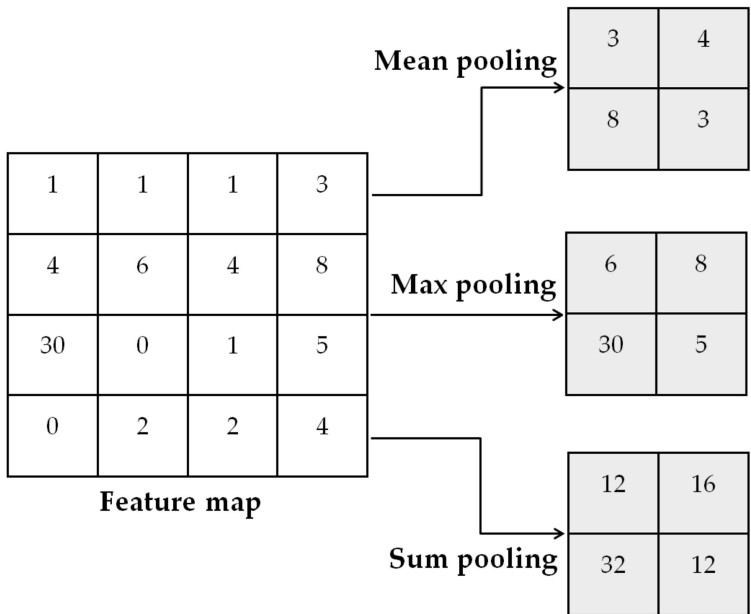

**Figure 4.** Examples of pooling using the mean, max and sum operation.

### 2.5. Classifier Layer

This layer appears in the CNN architecture after total convolutional and pooling layers; this is a fully connected layer that interprets the feature representations obtained by the previous layers and performs the high-level reasoning function. It has a similar principle to the conventional multilayer perceptron neural system, and in this layer, the CNN recognizes and classifies the images that are part of the output. In a multiclass classification problem, this fully connected layer has the same number of outputs as the classes defined in the study case to be solved. The Softmax function has become one of the most popular options for the classification task, due to its effectiveness [42].

## 3. Particle Swarm Optimization

It is a stochastic algorithm established on the intelligence of the swarm and inspired by the way birds forage for food; each bird is represented using particles which "move" in a multidimensional search space and "adjust" based on the experience of neighbors and your own.

The possible solution to the problem is depicted by the particle, which can be considered as "an individual element in a flock" [43]. PSO uses local and global information to find the best solution using a fitness function and the speeds at which the particles are moving.

PSO is very prone to premature convergence and falls into local optimum, so since its introduction in 1995 by Kennedy and Eberhart [44], various optimization variants have been proposed [45–48].

---

**Algorithm 1. The PSO algorithm**

---

Initialize the parameter of the problem (a random population).
**while** (completion criteria are not met)
**begin**
For each particle i do
**begin**
Update the position $p_i$ using (1).
Update the velocity $x_i$ using (2).
Evaluate the fitness value of the particle
If is necessary using (3)(4)
Update pbest$_i$(t) and gbest$_i$(t).
**end**
**end**

---

Algorithm 1 describes the process carried out by the PSO. This algorithm is defined by the equations that allow updating of the velocity with Equation (2) and the position with Equation (1).

$$p_i(t+1) = p_i(t) + x_i(t+1), \tag{1}$$

In Equation (1), $p_i(t)$ is the position of particle $i$ in a time $t$, within the search space. By adding a velocity $x_i(t)$ it is possible to change the position of the particle [45].

$$x_i(t+1) = x_i(t)\omega + c_1 r_1 [y_i - p_i(t)] + c_2 r_2 [\hat{y} - p_i(t)], \tag{2}$$

In Equation (2), $x$ represents the velocity and $i$ the particle. The parameters $c_1$ and $c_2$ define the cognitive and social factors, respectively. The random values in the interval [0,1] are depicted by $r_1$ and $r_2$, $\omega$ is an inertia weight and the best position of the particle (*pbest$_i$*) is determined by $y_i$ and the best global position (*gbest*) by $\hat{y}$.

The swarm is assumed to consist of $n$ particles, so an objective function $f$ is implemented to perform the computation of particle fitness with a maximization task. The personal and global best values are updated using Equations (3) and (4), respectively, at a time $t$ [48].

Thus, $i \in 1 \ldots n$

$$pbest_i(t+1) = \begin{cases} pbest_i(t) \; if \; f(pbest_i(t)) \leq f(p(t+1)) \\ p_i(t+1) \; if \; f(pbest_i(t)) > f(p_i(t+1)) \end{cases} \tag{3}$$

$$gbest(t+1) = max\{f(y), \; f(gbest(t))\} \\ where, \quad y \; \in \; pbest_0(t), \; pbest_1(t), \; \ldots, \; pbest_n(t) \tag{4}$$

According to Equations (1) and (2), the movements of the particle in the search space are illustrated in Figure 5.

The red and yellow circles represent the movement that a particle makes when the parameters $c_1$ and $c_2$ are updated. When $c_1 > c_2$, the particle moves in the direction of the yellow circle. When this condition is met, it means that the swarm performs the exploration process, so they "fly" in the search space to find the area that allows it to find the global optimum.

This movement allows the particles to perform long displacements, thus covering the whole search space. In the case of $c_2 > c_1$ then, the particle motion will be towards the red circle. It is here that the exploitation process takes place; it consists of the swarm "flying" in the best area of the search space, making small motions, which allow an intensive search [49].

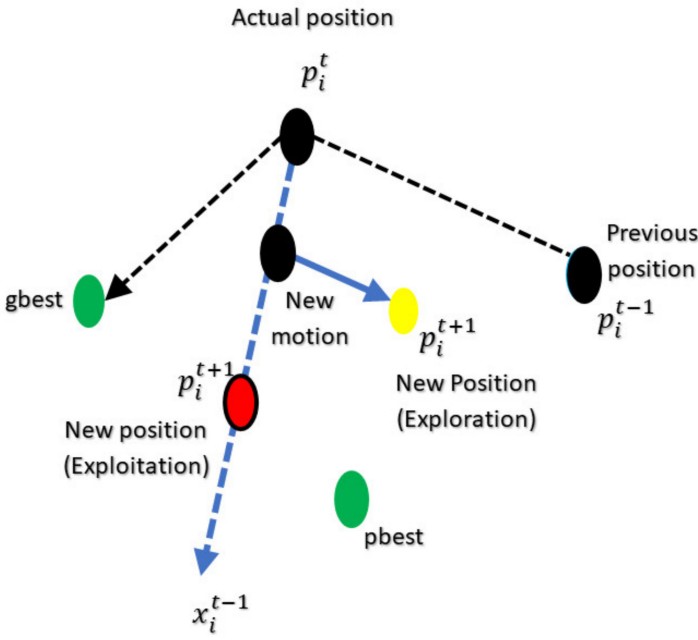

**Figure 5.** Representation of the movement of the particle.

## 4. Convolutional Neural Network Architecture Optimized by PSO

This Section presents two optimization approaches where the PSO algorithm is applied to optimize the parameters of CNN architectures, these approaches are denoted as PSO-CNN-I and PSO-CNN-II. The first objective is to select the most relevant parameters that have influence to obtain good performance of CNN and then implement the PSO algorithm to find these optimal parameters.

The parameters to be optimized were selected after evaluating the performance of a CNN with an experimental study, where the parameters were changed manually. As mentioned above, different CNN parameter values produce a variety of results for the same task; for this reason, the aim is to find the optimal architectures. The parameters listed below were chosen to be optimized in this work.

- The number of convolutional layers;
- The filter size or filter dimension used in each convolutional;
- The number of filters to extract the future maps (the convolution filter number);
- The batch size number: this value represents the number of images that are entered into CNN in each training block.

The general methodology of the proposal is presented in Figure 6, as the "training and optimization" block is the most important part of the whole process, where the CNN is initialized to integrate the parameter optimization by applying the PSO algorithm. In this process, the PSO is initialized according to the parameter given for the execution (the parameters are explained below) and this generates the particles. Each particle is a possible solution and its position has the parameter to be optimized, so each solution represents a complete CNN training.

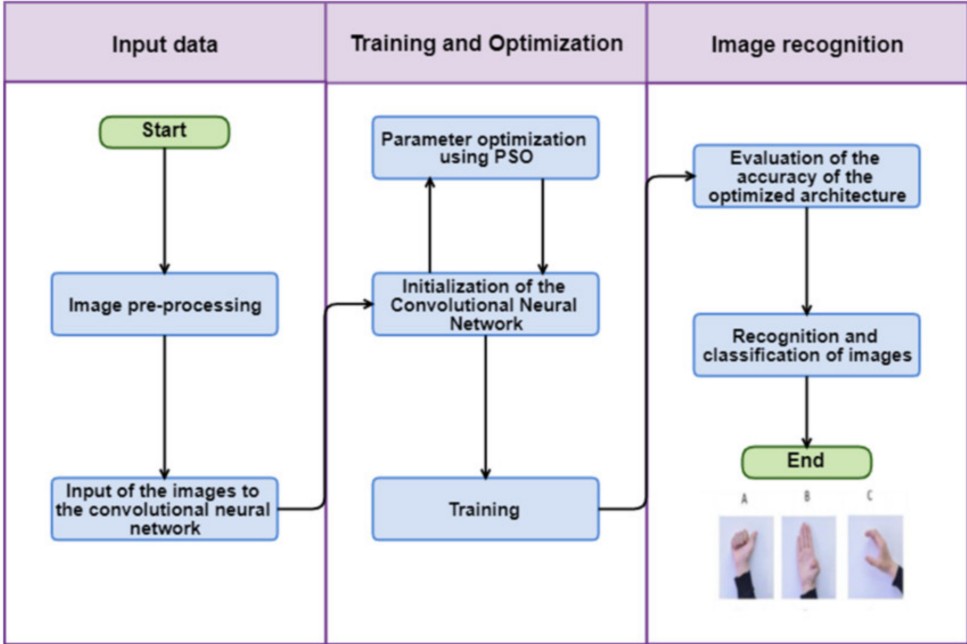

**Figure 6.** General CNN optimization process using the PSO algorithm. The letters A, B, and C stand the illustrated sign language in image form.

The training process is an iterative cycle that ends when all the particles generated by the PSO are evaluated for each generation. The computational cost is higher and, it depends on the database size, the size of particles, the number of iterations of the PSO and, the number of particles in each iteration. That is to say, if the PSO is executed with 10 particles and 10 iterations, the CNN training process is executed 100 times. The steps to optimize the CNN by the PSO algorithm are illustrated in Figure 7 and explained as follows.

1. Input database to train the CNN. This step consists of selecting the database to be processed and classified for the CNN (ASL alphabet, ASL MINIST and MSL alphabet). Is important to mention that all the elements of each database need to keep a similar structure or characteristics. In other words, images with the same scale and color gamma (grayscale, RGB, CMYK); additionally, with the same dimensions of pixels and a similar format of file (JPGE, PNG, TIFF, BMP, etc.).

2. Generate the particle population for the PSO algorithm. The PSO parameters are set to include the number of iterations, the number of particles, inertial weight, cognitive constant (W1), and social constant (W2); the parameters used in the experimentation are presented in Table 8. This step involves the design of the particles; the structures of these are presented in Tables 1 and 3 according to the two optimization architecture proposals in this paper.

3. Initialize the CNN architecture, with the parameter obtained by the PSO (convolution layers number, the filter size, number of convolution filters, and the batch size) the CNN is initialized and in conjunction with the additional parameter specified in Table 8, the CNN is ready to train the input database.

4. CNN training and validation. The CNN reads and processes the input databases taking the images for training, validation, and testing; this step produces a recognition rate and the AIC value. These values return to the PSO as part of the objective function.

5. Evaluate the objective function. The PSO algorithm evaluates the objective function to determine the best value. As in this research, we are considering two approaches, in the first, the objective function is only the recognition rate (Equation (5)) and in the second, the objective function consists of the recognition rate and the AIC value (Equation (6)).

6.  Update PSO parameters. At each iteration, each particle updates its velocity and position depending on its own best-known position (Pbest) in the search-space and the best-known position in the whole swarm (Gbest).
7.  The process is repeated, evaluating all the particles until the stop criteria are found (in this case, it is the number of iterations).
8.  Finally, the optimal solution is selected. In this process, the particle represented by Gbest is the optimal one for the CNN model.

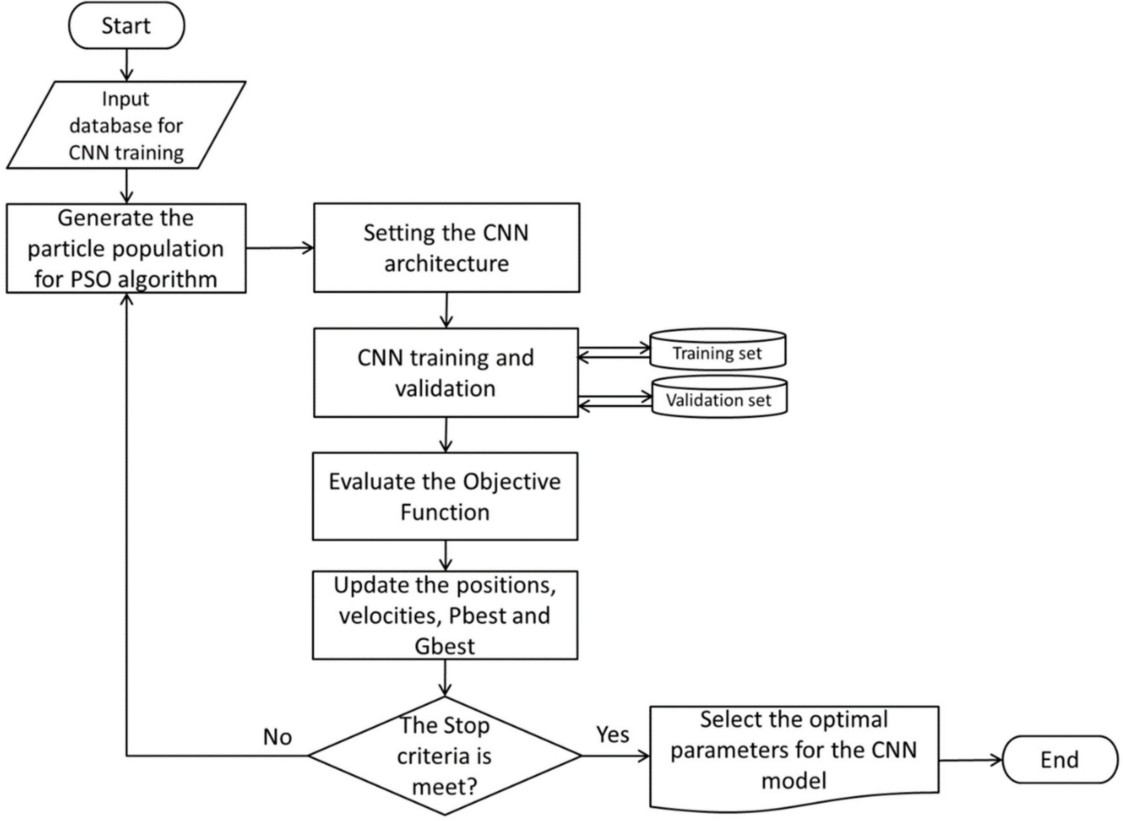

**Figure 7.** Flowchart of CNN optimization process using PSO.

### 4.1. PSO-CNN Optimization Process (PSO-CNN-I)

This first approach, which we are going to identify as PSO-CNN-I, consists of implementing a particle with four positions, one position for each parameter to be optimized (Figure 8). Table 1 presents the detail of the particle composition where the position $x_1$ corresponds to the number of layers with a search space from 1 to n, that is to say, that method can produce architectures with a minimum of one layer and maximum n, for the purposes of this work, we are using n = 3. The $x_2$ position represents the number of convolution filters used to extract the characteristics, with a search space of 32 to 18 filters. Position $x_3$ is the filter size; the search space is from 1 to 4 where this values represents a position, the value reached is mapped with the values of Table 2 to obtain the filter size (i.e., if the particle generates a value of 1 this represents a filter size of [3 × 3], to get a value of 2 the filter size will be [5 × 5] and so on, respectively, for each value. The last position represents the batch size ($x_4$), this is initialized considering the search space ranges from 32 to 256. In this optimization process, the consistency of the parameters between the layers is maintained in the same conditions, that is, if after the PSO execution it generates a particle with 3 convolutional layers ($x_1$), 50 filters ($x_2$), a filter dimension of 3 × 3 ($x_3$) and batch size of 50 ($x_4$). The same values of filter numbers ($x_2$) and filter size ($x_3$) will apply to the three convolution layers of the CNN.

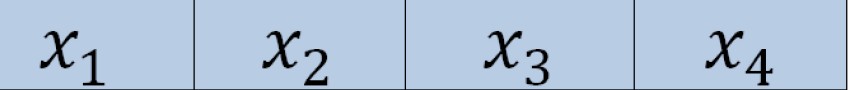

**Figure 8.** Structure of the particle used in the PSO-CNN-I approach.

**Table 1.** Search spaces used to define the particle in the PSO-CNN-I approach.

| Particle Coordinate | Hyper-Parameter | Search Space |
|:---:|:---|:---:|
| $x_1$ | Number of convolutional layers | [1, 3] |
| $x_2$ | Filter number | [32, 128] |
| $x_3$ | Filter size | [1, 4] |
| $x_4$ | Batch size in the training | [32, 256] |

**Table 2.** Convolutional filter dimensions for the $x_3$ position.

| $x_3$ Value | Search Space |
|:---:|:---:|
| 1 | [3, 3] |
| 2 | [5, 5] |
| 3 | [7, 7] |
| 4 | [9, 9] |

In this process, the objective function defined by Equation (5) is given by the recognition rate (precision) that the CNN returns after it is trained with the parameters generated by the PSO.

$$Objective\ function = Recognition\ Rate, \tag{5}$$

### 4.2. PSO-CNN Optimization Process (PSO-CNN-II)

In this second proposal, identifying as PSO-CNN-II, the particle structure consists of eight positions whose structure is presented in Figure 9, where each position represents the parameter to be optimized. The difference from the previous approach (PSO-CNN-I in Section 4.1) is finding more random searches in the architectures that the PSO produces; because in this case, the values for each convolution layer are completely different. Table 3 presents the detail of the particle composition, the description of each position, and the search space used. As we can see in Table 3, the positions $x_3$, $x_5$ and $x_7$ represent an index with an integer value between 1 to 4, and depending on the value taken by the PSO, a mapping is made with values presented in Table 2.

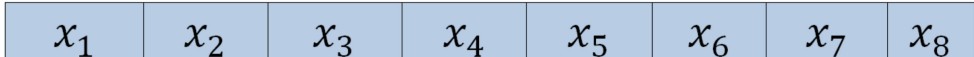

**Figure 9.** Structure of the particle used in the PSO-CNN-II approach.

**Table 3.** Search spaces used to define the particle in the PSO-CNN-II approach.

| Particle Coordinate | Hyper-Parameter | Search Space |
|:---:|:---|:---:|
| $x_1$ | Convolutional layer number | [1, 3] |
| $x_2$ | Filter number (layer 1) | [32, 128] |
| $x_3$ | Filter size (layer 1) | [1, 4] |
| $x_4$ | Filter number (layer 2) | [32, 128] |
| $x_5$ | Filter size (layer 2) | [1, 4] |
| $x_6$ | Filter number (layer 3) | [32, 128] |
| $x_7$ | Filter size (layer 3) | [1, 4] |
| $x_8$ | Batch size in the training | [32, 256] |

According to the values to optimize in this new approach, the $x_1$ position is used to control the number of convolution layers and the activation of the positions $x_2$ to $x_7$. If PSO generates a particle with a value of one for $x_1$, only the position $x_2$ and $x_3$ will be activated to generate the number of filters of the convolutional layer 1 and the filter size to use in this layer. In other words, if PSO produces a particle with a value of three in the $x_1$ position, the positions from $x_2$ to $x_7$ will be activated to generate the number of filters to use in the convolutional layer 1 ($x_2$), the filter size of layer 1 ($x_3$), the number of filters of layer 2 ($x_4$), the filter size for layer 2 ($x_5$), the number of filters of layer 3 ($x_6$), and the filter size for layer 3 ($x_7$), respectively; these values are completely different from each other, therefore this methodology helps to produce more heterogeneous CNN architectures.

Another difference between this proposal and the previous one (PSO-CNN-I) is that the objective function changes, for this we are using the recognition rate together with the Akaike information criteria (AIC). The AIC penalizes the architectures according to the number of parameters used; that is to say, the model is penalized when it needs more parameters. The objective function is considered the highest recognition rate and the lowest AIC. The AIC is defined in Equation (6).

$$AIC = 2k - 2ln(L), \tag{6}$$

According to our problem, in Equation (6) $k$ is the number of parameters of the model (number of layers and filter number) and $L$ is the maximum value of the recognition that the CNN can reach; in this case, the value is 100. Figure 10 illustrates an example of a particle generated by PSO.

| 3 | 100 | 1 | 85 | 2 | 50 | 3 | 32 |

**Figure 10.** Example of a particle generated by PSO.

Based on the structure of Figure 10, we have a three-layer convolutional architecture, where the first layer consists of 100 convolution filters with a filter size of $3 \times 3$. The second layer has 85 convolution filters with a filter size of $5 \times 5$ and the third convolution layer has 50 convolutional filters with a filter size of $7 \times 7$. Finally, the batch size is 32. The CNN is training with these values, and the recognition rate is calculated, additionally the AIC is obtained based on the parameters of the positions $x_1$, $x_2$, $x_4$ and $x_6$ which represents the number of convolution layers and the number of filters for each convolutional layer. After applying Equation (6), this architecture produces the AIC defined in Equation (7).

$$AIC = 2(3 + 100 + 85 + 50) - 2ln\,(100),$$
$$AIC = 466.7897, \tag{7}$$

Assuming that there are two architectures with the same recognition rate but with different AICs (Table 4), the model will take the architecture with the lowest AIC, as this would help penalize the parameters that are needed to train the network and thus produce optimized and simpler architectures.

**Table 4.** Objective function values based on the recognition rate and the AIC value.

| Architecture Number | Recognition Rate (%) | AIC Value |
|---|---|---|
| 1 | 98.50 | 466.78 |
| 2 | 98.50 | 350.85 |

## 5. Experiments and Results

This section describes the three databases implemented in the case studies (ASL alphabet, ASL MNIST, and MSL alphabet), the static parameters used to set the PSO algorithm and the CNN process, the experimental results obtained in the two optimization

approaches that were performed (PSO-CNN-I and PSO-CNN-II), as well as the comparison analysis against other approaches.

### 5.1. Sign Language Databases Used in the Study Cases

The characteristics of the sign databases are described below.

### 5.1.1. American Sign Language (ASL Alphabet)

The ASL alphabet consists of 87,000 images in color format, with a dimension of 200 × 200 pixels. This database contains 29 classes, these are labeled in a range of 0 to 28, with a one-to-one assignment for each letter of the American alphabet A–Z (0 to 25 for the alphabet; that is, 0 = A and 25 = Z) the other three classes correspond to the space symbols, delete, and null (26 to 28; i.e., 26 = space, 27 = delete and, 28 = null). Table 5 presents the general description of the ASL alphabet database and Figure 11 illustrates a sample of the images.

**Table 5.** ASL Alphabet database description.

| Name | ASL Alphabet Detail |
| --- | --- |
| Total images | 87,000 |
| Images for training | 82,650 |
| Images for test | 4350 |
| Images size | 32 × 32 |
| Database format | JPGE |

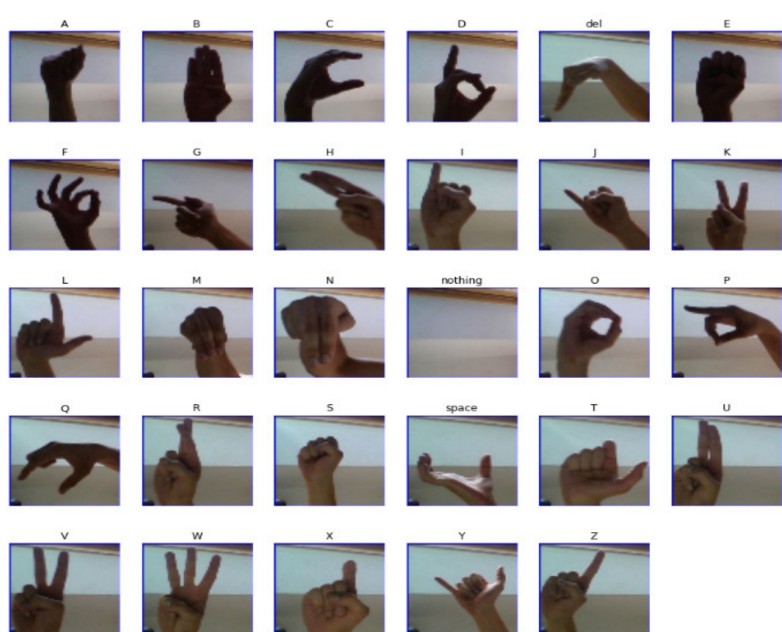

**Figure 11.** A sample of the ASL alphabet database.

### 5.1.2. American Sign Language (ASL MNIST)

ASL MNIST consists of a collection of 34,627 grayscale images with a dimension of 28 × 28 pixels. This database has 24 labeled classes in a range from 0 to 25 with assignment for each letter of the alphabet A–Z (the class 9 = J and 25 = Z, were excluded due to gestural movements). Table 6 presents a description of this database and Figure 12 illustrates a sample of the sign images.

**Table 6.** ASL MNIST database description.

| Name | ASL MNIST Detail |
|---|---|
| Total images | 34,627 |
| Images for training | 24,239 |
| Images for test | 10,388 |
| Images size | 28 × 28 |
| Database format | CSV |

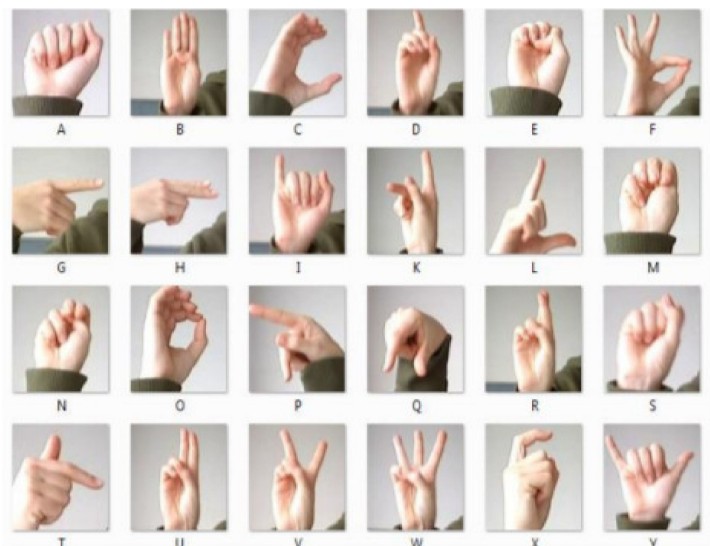

**Figure 12.** A sample of the ASL MNIST database.

### 5.1.3. Mexican Sign Language (MSL Alphabet)

The MSL alphabet database was obtained from a group of 18 people, including deaf students and sign language translation teachers. Students are part of an inclusive group in a high school in Mexico. This database consists of 21 classes with the alphabet of the MSL without movement as illustrated in Figure 13. Ten images were captured for each letter, achieving a total of 3780 grayscale images with a dimension of 32 by 32. Table 7 displays a general overview of the MSL alphabet database.

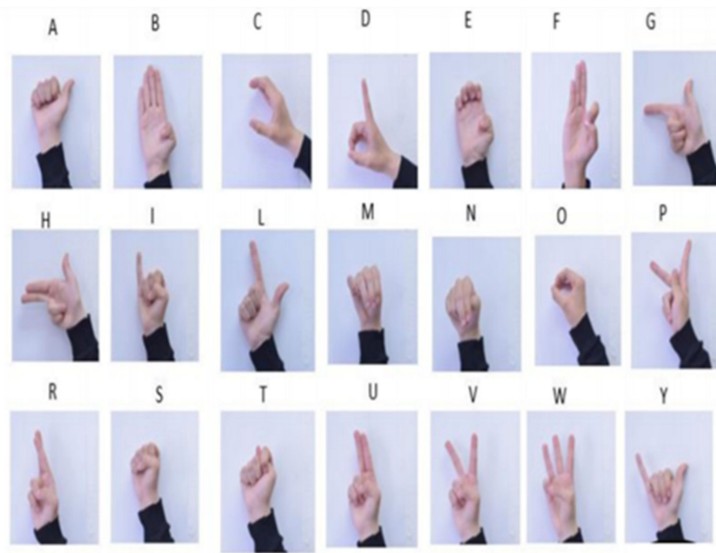

**Figure 13.** Sample of the MSL alphabet database.

**Table 7.** MSL alphabet database description.

| Name | MSL Alphabet Detail |
|---|---|
| Total images | 3780 |
| Images for training | 2646 |
| Images for test | 1134 |
| Images size | $32 \times 32$ |
| Database format | JPG |

### 5.2. Parameters Used in the Experimentation

In the CNN parameter settings, some static parameters were used, including the learning function, the activation function in the classifying layer, the non-linearity activation function, and the epoch number. The fixed parameters considered in the PSO configuration are the number of particles, the iterations number, the inertial weight, and the social and cognitive constants. The static parameters used for PSO and CNN are presented in Table 8. The dynamic parameters optimized by PSO are the number of convolutional layers, the size of the filters used in each convolutional layer, the number of convolutional filters, and the batch size (Tables 1 and 3).

**Table 8.** Static parameters for CNN and PSO.

| Parameters of CNN | |
|---|---|
| Learning function | Adam |
| Activation function (classifying layer) | Softmax |
| Non-linearity activation function | ReLU |
| Epochs | 5 |
| **Parameters of PSO** | |
| Particles | 10 |
| Iterations | 10 |
| Inertial weight (W) | 0.85 |
| Social constant (W2) | 2 |
| Cognitive constant (W1) | 2 |

### 5.3. Optimization Results Obtained by the PSO-CNN-I Approach

This section presents the simulation results produced after the CNN architecture is optimized considering the approach described in Section 4.1. The experimentation consists of 30 executions carried out on the three databases; the aim is to obtain the optimal CNN architecture, that is, minimum parameters necessary to maximize the recognition rate.

The first experiment was applied in the ASL alphabet (Table 5), using a distribution of 80% of the total images for training and 20% for testing. Table 9 shows the values achieved after 30 executions, where the higher recognition rate was a value of 99.87% and the mean was 99.58%. Based on the results, we can see that the optimal architecture achieved by the PSO was as follows: three convolutional layers, 128 filters per layer with a filter size of $7 \times 7$, and the batch size with a value of 256.

In another test, the PSO-CNN-I approach was applied to the ASL MNIST database; Table 10 presents the results achieved by the CNN where the best recognition rate was a value of 98.82% and the mean of 99.53%. According to this analysis, the optimal architecture for this study case is two convolutional layers, with 117 convolutional filters in both layers with a filter size of $7 \times 7$ and the batch size with a value of 129.

**Table 9.** Results achieved by the PSO-CNN-I in ASL alphabet database.

| No. | No. Layers | No. Filters | Filter Size | Batch Size | Recognition Rate (%) |
|-----|-----|-----|-----|-----|-----|
| 1 | 3 | 99 | [7 × 7] | 107 | 98.85 |
| 2 | 3 | 104 | [9 × 9] | 256 | 99.66 |
| 3 | 3 | 128 | [9 × 9] | 256 | 99.70 |
| 4 | 3 | 128 | [7 × 7] | 256 | 99.79 |
| 5 | 3 | 128 | [9 × 9] | 256 | 99.72 |
| 6 | 3 | 128 | [7 × 7] | 256 | 99.62 |
| 7 | 2 | 32 | [7 × 7] | 256 | 98.18 |
| 8 | 3 | 109 | [7 × 7] | 256 | 99.73 |
| 9 | 3 | 128 | [7 × 7] | 197 | 99.75 |
| 10 | 3 | 128 | [7 × 7] | 256 | 99.81 |
| 11 | 3 | 66 | [7 × 7] | 181 | 99.31 |
| 12 | 3 | 118 | [7 × 7] | 256 | 99.87 |
| 13 | 3 | 128 | [9 × 9] | 256 | 99.67 |
| 14 | 3 | 128 | [7 × 7] | 256 | 99.85 |
| 15 | 3 | 128 | [9 × 9] | 256 | 99.61 |
| 16 | 3 | 128 | [9 × 9] | 256 | 99.63 |
| 17 | 3 | 90 | [9 × 9] | 256 | 99.66 |
| 18 | 3 | 128 | [7 × 7] | 256 | 99.82 |
| 19 | 3 | 128 | [7 × 7] | 256 | 99.79 |
| 20 | 3 | 128 | [7 × 7] | 256 | 99.76 |
| 21 | 3 | 128 | [9 × 9] | 256 | 99.68 |
| 22 | 3 | 128 | [9 × 9] | 256 | 99.67 |
| 23 | 3 | 128 | [7 × 7] | 256 | 99.75 |
| 24 | 3 | 123 | [7 × 7] | 32 | 98.38 |
| 25 | 3 | 128 | [9 × 9] | 256 | 99.64 |
| 26 | 3 | 128 | [7 × 7] | 256 | 99.82 |
| 27 | 3 | 128 | [9 × 9] | 215 | 99.56 |
| 28 | 3 | 128 | [7 × 7] | 256 | 99.87 |
| 29 | 3 | 100 | [9 × 9] | 256 | 99.64 |
| 30 | 3 | 128 | [7 × 7] | 256 | 99.84 |
| | | | | Mean | 99.58 |

**Table 10.** Results achieved by the PSO-CNN-I in ASL MNIST database.

| No. | No. Layers | No. Filters | Filter Size | Batch Size | Recognition Rate (%) |
|-----|-----|-----|-----|-----|-----|
| 1 | 3 | 128 | [9 × 9] | 137 | 99.27 |
| 2 | 2 | 128 | [9 × 9] | 218 | 99.54 |
| 3 | 2 | 128 | [7 × 7] | 205 | 99.52 |
| 4 | 3 | 128 | [7 × 7] | 136 | 99.33 |
| 5 | 2 | 128 | [9 × 9] | 232 | 99.59 |
| 6 | 3 | 96 | [9 × 9] | 107 | 98.82 |
| 7 | 2 | 118 | [7 × 7] | 189 | 99.36 |
| 8 | 2 | 128 | [9 × 9] | 256 | 99.59 |
| 9 | 2 | 112 | [9 × 9] | 256 | 99.49 |
| 10 | 2 | 128 | [9 × 9] | 256 | 99.60 |
| 11 | 2 | 128 | [7 × 7] | 256 | 99.59 |
| 12 | 2 | 128 | [7 × 7] | 256 | 99.61 |
| 13 | 2 | 128 | [9 × 9] | 220 | 99.67 |
| 14 | 2 | 128 | [9 × 9] | 256 | 99.57 |
| 15 | 2 | 128 | [9 × 9] | 256 | 99.51 |
| 16 | 2 | 128 | [7 × 7] | 237 | 99.55 |
| 17 | 2 | 128 | [7 × 7] | 256 | 99.61 |

**Table 10.** *Cont.*

| No. | No. Layers | No. Filters | Filter Size | Batch Size | Recognition Rate (%) |
|---|---|---|---|---|---|
| 18 | 2 | 128 | [9 × 9] | 256 | 99.58 |
| 19 | 2 | 128 | [9 × 9] | 256 | 99.53 |
| 20 | 2 | 128 | [9 × 9] | 256 | 99.65 |
| 21 | 2 | 128 | [7 × 7] | 148 | 99.42 |
| 22 | 2 | 128 | [9 × 9] | 256 | 99.51 |
| 23 | 2 | 128 | [9 × 9] | 215 | 99.53 |
| 24 | 2 | 128 | [9 × 9] | 255 | 99.56 |
| 25 | 2 | 128 | [9 × 9] | 256 | 99.65 |
| 26 | 2 | 128 | [7 × 7] | 256 | 99.57 |
| 27 | 2 | 128 | [9 × 9] | 256 | 99.53 |
| 28 | 2 | 117 | [7 × 7] | 129 | 99.98 |
| 29 | 3 | 128 | [5 × 5] | 242 | 99.87 |
| 30 | 2 | 128 | [7 × 7] | 256 | 99.55 |
| | | | | Mean | 99.53 |

Table 11 presents the experimental results obtained when the approach is applied in the MSL alphabet database. As we can see in Table 11, the best accuracy reached by the CNN was 99.37% with a mean of 99.10%. In this case, the optimal architecture is as follows: one convolutional layer with 122 convolutional filters, filter size of 3 × 3, and batch size of 128.

**Table 11.** Results achieved by the PSO-CNN-I in MSL alphabet database.

| No. | No. Layers | No. Filters | Filter Size | Batch Size | Recognition Rate (%) |
|---|---|---|---|---|---|
| 1 | 2 | 101 | [7 × 7] | 93 | 98.95 |
| 2 | 1 | 128 | [3 × 3] | 56 | 98.95 |
| 3 | 1 | 110 | [3 × 3] | 52 | 98.82 |
| 4 | 1 | 128 | [3 × 3] | 121 | 99.20 |
| 5 | 1 | 128 | [3 × 3] | 128 | 99.32 |
| 6 | 1 | 128 | [3 × 3] | 128 | 99.07 |
| 7 | 1 | 128 | [3 × 3] | 110 | 99.24 |
| 8 | 1 | 101 | [5 × 5] | 114 | 98.82 |
| 9 | 1 | 128 | [3 × 3] | 128 | 99.24 |
| 10 | 1 | 74 | [3 × 3] | 88 | 98.95 |
| 11 | 1 | 128 | [3 × 3] | 128 | 99.32 |
| 12 | 1 | 128 | [3 × 3] | 32 | 98.48 |
| 13 | 1 | 128 | [3 × 3] | 93 | 99.28 |
| 14 | 1 | 128 | [3 × 3] | 97 | 99.11 |
| 15 | 1 | 128 | [3 × 3] | 32 | 98.74 |
| 16 | 1 | 128 | [3 × 3] | 72 | 99.32 |
| 17 | 1 | 128 | [3 × 3] | 93 | 99.37 |
| 18 | 1 | 63 | [3 × 3] | 47 | 98.44 |
| 19 | 1 | 128 | [3 × 3] | 128 | 99.20 |
| 20 | 1 | 126 | [3 × 3] | 128 | 99.28 |
| 21 | 1 | 128 | [3 × 3] | 128 | 99.32 |
| 22 | 1 | 128 | [3 × 3] | 83 | 99.20 |
| 23 | 1 | 128 | [3 × 3] | 63 | 99.20 |
| 24 | 1 | 122 | [3 × 3] | 128 | 99.37 |
| 25 | 1 | 128 | [3 × 3] | 128 | 99.32 |
| 26 | 1 | 114 | [3 × 3] | 84 | 99.32 |
| 27 | 1 | 128 | [3 × 3] | 32 | 98.74 |
| 28 | 1 | 128 | [3 × 3] | 89 | 99.28 |
| 29 | 1 | 43 | [3 × 3] | 53 | 97.81 |
| 30 | 1 | 128 | [3 × 3] | 72 | 98.99 |
| | | | | Mean | 99.10 |

### 5.4. Optimization Results Obtained by the PSO-CNN-II Approach

The results presented in this section consist of 30 executions of the PSO-CNN-II approach applied in the ASL alphabet, ASL MNIST, and MSL alphabet databases; the objective is to maximize the recognition rate and minimize the value of AIC.

The experimental results obtained from the ASL alphabet database after applying the PSO-CNN-II optimization approach (Section 4.2) are presented in Table 12. In this test, the database was distributed so that 70% of the data were kept for the training phase and 30% of the data for testing. Table 12 shows the best recognition rate with a value of 99.23% and a mean of 98.69%. The best architecture found by the PSO for the CNN had the following structure: three convolutional layers where the first layer had 84 convolutional filters and $3 \times 3$ size filters; the second layer with 128 convolutional filters with the size of $9 \times 9$ and the third layer with 128 convolutional filters and $7 \times 7$ size filters. In this approach, the objective function is composed of the recognition rate and the AIC value; that is, the best recognition rate is evaluated first and then the AIC value, if it was the case that CNN achieved two or more architectures with the same recognition rate, the process takes the architecture with the minimum AIC, with the goal of achieving an optimal architecture with the fewest parameters and the highest recognition rate.

**Table 12.** Results achieved by the PSO-CNN-II in ASL alphabet database.

| No. | No. Layers | Layer 1 | | Layer 2 | | Layer 3 | | Batch Size | AIC Value | (%) Recogn. Rate |
| --- | --- | --- | --- | --- | --- | --- | --- | --- | --- | --- |
| | | No. Filters | Filter Size | No. Filters | Filter Size | No. Filters | Filter Size | | | |
| 1 | 3 | 128 | [3 × 3] | 128 | [7 × 7] | 128 | [5 × 5] | 256 | 764.78 | 98.99 |
| 2 | 3 | 128 | [5 × 5] | 121 | [5 × 5] | 128 | [5 × 5] | 213 | 750.78 | 98.73 |
| 3 | 3 | 84 | [3 × 3] | 128 | [7 × 7] | 128 | [5 × 5] | 84 | 676.78 | 99.23 |
| 4 | 2 | 45 | [5 × 5] | 128 | [7 × 7] | 0 | 0 | 0 | 340.78 | 98.15 |
| 5 | 3 | 32 | [3 × 3] | 128 | [7 × 7] | 128 | [3 × 3] | 256 | 572.78 | 98.86 |
| 6 | 3 | 128 | [3 × 3] | 128 | [7 × 7] | 128 | [5 × 5] | 256 | 764.78 | 98.96 |
| 7 | 3 | 84 | [5 × 5] | 128 | [5 × 5] | 128 | [3 × 3] | 256 | 676.78 | 98.85 |
| 8 | 3 | 128 | [3 × 3] | 128 | [7 × 7] | 128 | [5 × 5] | 256 | 764.78 | 99.02 |
| 9 | 3 | 32 | [5 × 5] | 128 | [5 × 5] | 128 | [3 × 3] | 256 | 572.78 | 98.9 |
| 10 | 3 | 124 | [3 × 3] | 128 | [7 × 7] | 128 | [3 × 3] | 256 | 756.78 | 98.64 |
| 11 | 3 | 32 | [3 × 3] | 128 | [7 × 7] | 128 | [7 × 7] | 256 | 572.78 | 98.93 |
| 12 | 3 | 32 | [3 × 3] | 128 | [7 × 7] | 128 | [3 × 3] | 256 | 572.78 | 98.53 |
| 13 | 3 | 128 | [3 × 3] | 128 | [7 × 7] | 128 | [5 × 5] | 256 | 764.78 | 99.01 |
| 14 | 3 | 73 | [7 × 7] | 128 | [7 × 7] | 108 | [3 × 3] | 256 | 614.78 | 97.91 |
| 15 | 3 | 128 | [3 × 3] | 128 | [7 × 7] | 128 | [5 × 5] | 256 | 764.78 | 99.06 |
| 16 | 2 | 128 | [3 × 3] | 128 | [7 × 7] | 0 | 0 | 0 | 506.78 | 98.23 |
| 17 | 2 | 88 | [7 × 7] | 128 | [7 × 7] | 0 | 0 | 0 | 426.78 | 97.4 |
| 18 | 3 | 32 | [5 × 5] | 128 | [7 × 7] | 128 | [5 × 5] | 256 | 572.78 | 99.06 |
| 19 | 2 | 128 | [5 × 5] | 119 | [7 × 7] | 0 | 0 | 0 | 488.78 | 98.1 |
| 20 | 3 | 116 | [3 × 3] | 128 | [5 × 5] | 128 | [7 × 7] | 252 | 740.78 | 98.96 |
| 21 | 3 | 49 | [5 × 5] | 128 | [7 × 7] | 128 | [7 × 7] | 256 | 606.78 | 98.93 |
| 22 | 2 | 128 | [3 × 3] | 128 | [7 × 7] | 0 | 0 | 0 | 506.78 | 98.19 |
| 23 | 3 | 32 | [5 × 5] | 128 | [7 × 7] | 128 | [5 × 5] | 256 | 572.78 | 98.96 |
| 24 | 2 | 32 | [5 × 5] | 128 | [7 × 7] | 0 | 0 | 0 | 314.78 | 98.04 |
| 25 | 2 | 128 | [5 × 5] | 81 | [5 × 5] | 0 | 0 | 0 | 412.78 | 98.92 |
| 26 | 3 | 32 | [5 × 5] | 128 | [7 × 7] | 128 | [3 × 3] | 256 | 572.78 | 98.58 |
| 27 | 3 | 32 | [3 × 3] | 128 | [7 × 7] | 128 | [5 × 5] | 256 | 572.78 | 98.88 |
| 28 | 3 | 128 | [3 × 3] | 128 | [7 × 7] | 128 | [5 × 5] | 256 | 764.78 | 99.02 |
| 29 | 3 | 128 | [3 × 3] | 128 | [7 × 7] | 128 | [3 × 3] | 256 | 764.78 | 99.08 |
| 30 | 3 | 128 | [3 × 3] | 128 | [7 × 7] | 128 | [3 × 3] | 256 | 764.78 | 98.71 |
| | | | | | | | | | Mean | 98.69 |

In Table 13, we present the results where the PSO-CNN-II was implemented in the ASL MNIST database. In this test, the best recognition rate was 99.80%, an AIC value of 506.79 and, a mean of 99.48%. The optimal parameters found by the PSO were the

following: two-layer CNN architecture, the first layer had 128 filters of convolution and a filter size of 5 × 5; the second layer had 128 convolutional filters with a filter size of 9 × 9, and the batch size was 128.

**Table 13.** Results achieved by the PSO-CNN-II in ASL MNIST database.

| No. | No. Layers | Layer 1 | | Layer 2 | | Layer 3 | | Batch Size | AIC Value | (%) Recogn. Rate |
| | | No. Filters | Filter Size | No. Filters | Filter Size | No. Filters | Filter Size | | | |
|---|---|---|---|---|---|---|---|---|---|---|
| 1 | 2 | 128 | [5 × 5] | 128 | [9 × 9] | 0 | 0 | 128 | 506.79 | 99.80 |
| 2 | 2 | 74 | [9 × 9] | 114 | [9 × 9] | 0 | 0 | 174 | 370.79 | 99.42 |
| 3 | 3 | 32 | [5 × 5] | 128 | [9 × 9] | 128 | [5 × 5] | 122 | 572.79 | 99.53 |
| 4 | 2 | 125 | [5 × 5] | 125 | [9 × 9] | 0 | 0 | 147 | 503.79 | 99.58 |
| 5 | 2 | 90 | [5 × 5] | 128 | [9 × 9] | 0 | 0 | 256 | 500.79 | 99.68 |
| 6 | 3 | 32 | [3 × 3] | 128 | [9 × 9] | 128 | [9 × 9] | 148 | 572.79 | 99.51 |
| 7 | 2 | 121 | [7 × 7] | 95 | [9 × 9] | 0 | 0 | 100 | 426.79 | 99.26 |
| 8 | 3 | 32 | [7 × 7] | 128 | [9 × 9] | 125 | [9 × 9] | 256 | 569.79 | 99.6 |
| 9 | 2 | 32 | [9 × 9] | 126 | [9 × 9] | 0 | 0 | 106 | 310.79 | 99.4 |
| 10 | 3 | 115 | [7 × 7] | 102 | [9 × 9] | 128 | [7 × 7] | 215 | 686.79 | 99.42 |
| 11 | 2 | 32 | [9 × 9] | 128 | [9 × 9] | 0 | 0 | 256 | 314.79 | 99.44 |
| 12 | 2 | 77 | [7 × 7] | 100 | [9 × 9] | 0 | 0 | 183 | 348.79 | 99.59 |
| 13 | 2 | 87 | [7 × 7] | 128 | [9 × 9] | 0 | 0 | 256 | 424.79 | 99.7 |
| 14 | 2 | 32 | [9 × 9] | 128 | [9 × 9] | 0 | 0 | 256 | 314.79 | 99.53 |
| 15 | 3 | 32 | [5 × 5] | 103 | [9 × 9] | 125 | [9 × 9] | 256 | 516.79 | 99.53 |
| 16 | 2 | 70 | [9 × 9] | 126 | [9 × 9] | 0 | 0 | 256 | 386.79 | 99.63 |
| 17 | 2 | 64 | [7 × 7] | 128 | [9 × 9] | 0 | 0 | 256 | 378.79 | 99.7 |
| 18 | 3 | 32 | [7 × 7] | 77 | [9 × 9] | 128 | [9 × 9] | 256 | 470.79 | 99.36 |
| 19 | 2 | 128 | [7 × 7] | 128 | [9 × 9] | 0 | 0 | 256 | 506.79 | 99.74 |
| 20 | 3 | 32 | [3 × 3] | 128 | [9 × 9] | 128 | [5 × 5] | 32 | 572.79 | 98.95 |
| 21 | 3 | 32 | [7 × 7] | 128 | [9 × 9] | 123 | [7 × 7] | 162 | 577.79 | 99.33 |
| 22 | 2 | 51 | [9 × 9] | 128 | [9 × 9] | 0 | 0 | 194 | 352.79 | 99.47 |
| 23 | 2 | 50 | [7 × 7] | 128 | [9 × 9] | 0 | 0 | 256 | 350.79 | 99.63 |
| 24 | 2 | 128 | [7 × 7] | 128 | [9 × 9] | 0 | 0 | 162 | 506.79 | 99.67 |
| 25 | 2 | 100 | [5 × 5] | 76 | [5 × 5] | 0 | 0 | 76 | 346.79 | 98.23 |
| 26 | 2 | 52 | [9 × 9] | 128 | [7 × 7] | 0 | 0 | 256 | 354.79 | 99.54 |
| 27 | 2 | 128 | [5 × 5] | 128 | [9 × 9] | 0 | 0 | 142 | 506.79 | 99.53 |
| 28 | 3 | 83 | [3 × 3] | 125 | [9 × 9] | 0 | 0 | 136 | 410.79 | 99.38 |
| 29 | 3 | 128 | [5 × 5] | 128 | [9 × 9] | 128 | [9 × 9] | 256 | 764.79 | 99.57 |
| 30 | 2 | 74 | [7 × 7] | 120 | [9 × 9] | 0 | 0 | 256 | 382.79 | 99.72 |
| | | | | | | | | | Mean | 99.48 |

In another experiment, the optimization approach was applied to the MSL alphabet database after 30 simulations. The results obtained are presented in Table 14, where the best recognition rate was 99.45% with an AIC of 248.79. The general mean for this study case was a value of 98.91%. In this optimization, one-layer CNN architecture was achieved, with 128 convolutional filters, 3 × 3 filter sizes, and 154 batch sizes.

*5.5. Statistical Test between PSO-CNN-I and PSO-CNN-II Optimization Process*

Table 15 presents a summary of the results obtained after the two approaches were applied to the three databases. We can see that good results were achieved in all the cases; we can analyze that for the ASL alphabet and the ASL MNIST, the PSO-CNN-I optimization approach was better with mean values of 99.58% and 99.53%, respectively. For the MSL alphabet database, the PSO-CNN-II optimization method achieved a better recognition rate with a mean value of 98.91%. Although, if the results were analyzed with respect to the AIC value, for the ASL MNIST and the MSL alphabet, the PSO-CNN-I reached the lowest values with AIC of 462.79 and 236.80, respectively, and for ASL alphabet, the PSO-CNN-II achieved a better AIC value. A low AIC value means that the CNN architecture required fewer parameters, so it is important to determine what is most relevant to any problem,

the CNN accuracy, or to configure the CNN architectures with minimal parameters that can be implemented in real-time systems.

**Table 14.** Results achieved by the PSO-CNN-II in MSL alphabet.

| No. | No. Layers | Layer 1 | | Layer 2 | | Layer 3 | | BATCH SIZE | AIC Value | (%) Recogn. Rate |
|---|---|---|---|---|---|---|---|---|---|---|
| | | No. Filters | Filter Size | No. Filters | Filter Size | No. Filters | Filter Size | | | |
| 1 | 1 | 128 | [3 × 3] | 0 | 0 | 0 | 0 | 32 | 248.79 | 98.74 |
| 2 | 1 | 128 | [3 × 3] | 0 | 0 | 0 | 0 | 163 | 248.79 | 99.28 |
| 3 | 1 | 116 | [3 × 3] | 0 | 0 | 0 | 0 | 105 | 224.79 | 98.99 |
| 4 | 1 | 81 | [3 × 3] | 0 | 0 | 0 | 0 | 32 | 154.79 | 98.44 |
| 5 | 1 | 128 | [3 × 3] | 0 | 0 | 0 | 0 | 149 | 248.79 | 98.90 |
| 6 | 1 | 128 | [3 × 3] | 0 | 0 | 0 | 0 | 221 | 248.79 | 98.57 |
| 7 | 1 | 128 | [3 × 3] | 0 | 0 | 0 | 0 | 57 | 248.79 | 99.24 |
| 8 | 1 | 67 | [3 × 3] | 0 | 0 | 0 | 0 | 246 | 126.73 | 97.77 |
| 9 | 1 | 118 | [3 × 3] | 0 | 0 | 0 | 0 | 113 | 228.79 | 99.16 |
| 10 | 1 | 128 | [3 × 3] | 0 | 0 | 0 | 0 | 154 | 248.79 | 99.45 |
| 11 | 1 | 103 | [3 × 3] | 0 | 0 | 0 | 0 | 92 | 198.79 | 99.03 |
| 12 | 1 | 65 | [3 × 3] | 0 | 0 | 0 | 0 | 32 | 122.79 | 98.32 |
| 13 | 1 | 128 | [3 × 3] | 0 | 0 | 0 | 0 | 94 | 248.79 | 99.07 |
| 14 | 1 | 128 | [3 × 3] | 0 | 0 | 0 | 0 | 90 | 248.79 | 99.11 |
| 15 | 1 | 112 | [3 × 3] | 0 | 0 | 0 | 0 | 97 | 216.79 | 99.24 |
| 16 | 1 | 128 | [3 × 3] | 0 | 0 | 0 | 0 | 32 | 248.79 | 98.74 |
| 17 | 1 | 128 | [3 × 3] | 0 | 0 | 0 | 0 | 46 | 248.79 | 98.65 |
| 18 | 1 | 128 | [3 × 3] | 0 | 0 | 0 | 0 | 199 | 248.79 | 98.32 |
| 19 | 1 | 128 | [3 × 3] | 0 | 0 | 0 | 0 | 244 | 248.79 | 99.03 |
| 20 | 1 | 120 | [3 × 3] | 0 | 0 | 0 | 0 | 32 | 232.79 | 99.07 |
| 21 | 1 | 128 | [3 × 3] | 0 | 0 | 0 | 0 | 105 | 248.79 | 99.16 |
| 22 | 1 | 128 | [3 × 3] | 0 | 0 | 0 | 0 | 77 | 248.79 | 99.03 |
| 23 | 1 | 108 | [3 × 3] | 0 | 0 | 0 | 0 | 84 | 208.79 | 99.07 |
| 24 | 1 | 54 | [3 × 3] | 0 | 0 | 0 | 0 | 32 | 100.79 | 98.44 |
| 25 | 1 | 102 | [3 × 3] | 0 | 0 | 0 | 0 | 102 | 196.79 | 99.03 |
| 26 | 1 | 128 | [3 × 3] | 0 | 0 | 0 | 0 | 114 | 248.79 | 99.20 |
| 27 | 1 | 119 | [3 × 3] | 0 | 0 | 0 | 0 | 256 | 230.79 | 98.61 |
| 28 | 1 | 98 | [3 × 3] | 0 | 0 | 0 | 0 | 122 | 188.79 | 99.20 |
| 29 | 1 | 128 | [3 × 3] | 0 | 0 | 0 | 0 | 83 | 248.79 | 99.07 |
| 30 | 1 | 128 | [3 × 3] | 0 | 0 | 0 | 0 | 135 | 248.79 | 99.37 |
| | | | | | | | | | Mean | 98.91 |

**Table 15.** Summary of the results obtained in the PSO-CNN-I and PSO-CNN-II approaches.

| Database | PSO-CNN-I | | | PSO-CNN-II | | |
|---|---|---|---|---|---|---|
| | Best | Mean | AIC | Best | Mean | AIC |
| ASL alphabet | 99.87% | 99.58% | 764.79 | 99.23% | 98.69% | 676.78 |
| ASL MNIST | 99.98% | 99.53% | 462.79 | 99.80% | 99.48% | 506.79 |
| MSL alphabet | 99.37% | 99.05% | 236.80 | 99.45% | 98.91% | 248.79 |

To confirm if significant evidence exists between the architectures and to identify which is better, the Wilcoxon signed-rank test was applied [50]; this is a non-parametric test that is recommended to be applied when the numerical data are not normally distributed, as is the case with the experimental results of metaheuristic algorithms. The Wilcoxon test was performed to compare the PSO-CNN-I and PSO-CNN-II optimization processes, considering the results presented in Tables 9–14. The general description of the values used to execute the Wilcoxon test is presented in Table 16 and described below:

- A confidence level of 95% ($\alpha = 0.05$).
- The null hypothesis is given that ($H_0$): the PSO-CNN-I architecture ($\mu_1$) is equal to PSO-CNN-II architecture ($\mu_2$), expressed as $H_0 : \mu_1 = \mu_2$.
- The alternative hypothesis is ($H_1$): affirm that PSO-CNN-I architecture ($\mu_1$) is greater than that PSO-CNN-II architecture ($\mu_2$), expressed as $H_1 : \mu_1 > \mu_2$ (Affirmation).
- The objective is to reject the hypothesis null ($H_0$) and support the alternative hypothesis ($H_1$).

**Table 16.** General description of the Wilcoxon test.

| | Description | Hypothesis |
|---|---|---|
| Null hypothesis | PSO-CNN-I architecture ($\mu_1$) = PSO-CNN-II architecture ($\mu_2$) | $H_0 : \mu_1 = \mu_2$ |
| Alternative hypothesis | PSO-CNN-I architecture ($\mu_1$) > PSO-CNN-II architecture ($\mu_2$), | $H_1 : \mu_1 > \mu_2$ (Affirmation) |

The first Wilcoxon test was applied for the ASL alphabet results (Tables 9 and 12). Table 17 shows the $R^+$, $R^-$, and the p-value (the p-values have been computed by using SPSS), where $R^+$ represents the sum of ranks for the problems in which the first algorithm outperformed the second, and $R-$ the sum of ranks for the opposite. The results obtained indicate an $R^+$ of 455, an $R^-$ of 10 and the *p*-value of <0.001. Because the *p*-value is less than the alpha value of $\alpha = 0.05$, then we support the alternative hypothesis with a 95% level of evidence, and we can affirm that the PSO-CNN-I architecture is better than the PSO-CNN-II.

**Table 17.** Wilcoxon test results for the ASL alphabet.

| Comparison PSO-CNN-I ($\mu_1$)—PSO-CNN-II ($\mu_2$) | $R^+$ | $R^-$ | *p*-Value |
|---|---|---|---|
| ASL alphabet | 455 | 10 | <0.001 |

Table 18 presents the results after the Wilcoxon test was applied for the ASL MNIST results (Tables 10 and 13). This test obtains the values $R^+ = 245.5$, $R^- = 189.5$, and the *p*-value = 0.545. Since the *p*-value is greater than the alpha value of $\alpha = 0.05$, the null hypothesis is accepted with a 95% level of evidence; therefore, we can affirm that evidence does not exist to determine that the PSO-CNN-I architecture is better than the PSO-CNN-II.

**Table 18.** Wilcoxon test results for the ASL MNIST.

| Comparison PSO-CNN-I ($\mu_1$)—PSO-CNN-II ($\mu_2$) | $R^+$ | $R^-$ | *p*-Value |
|---|---|---|---|
| ASL MNIST | 245.5 | 189.5 | 0.545 |

Finally, Table 19 presents the Wilcoxon test for the results of the MSL alphabet (Tables 11 and 14). The results obtained indicate the values $R^+ = 291$, $R^- = 115$, and the *p*-value = 0.045. We can see that the *p*-value is less than the alpha value of $\alpha = 0.05$; therefore, we support the alternative hypothesis with a level of evidence of 95%, and we can affirm that the PSO-CNN-I architecture is better than the PSO-CNN-II.

**Table 19.** Z-test results for the ASL MNIST.

| Comparison PSO-CNN-I ($\mu_1$)—PSO-CNN-II ($\mu_2$) | $R^+$ | $R^-$ | *p*-Value |
|---|---|---|---|
| MSL alphabet | 291 | 115 | 0.045 |

*5.6. State-of-the-Art Analysis Comparison*

To obtain more evidence about the performance of the optimization approaches presented in this paper, we make a comparative analysis (Table 20) against the state-of-art research, where CNN models are implemented in Alphabet Sign Language database recognition. The results presented in Table 20 represent the best recognition rate values reported by the authors, the detail of which is explained as follows: Zhao et al. [51] reports an accuracy of 89.32%, the CNN architecture has two convolutional layers, two pooling layers, the batch size is 150, and 80 iterations. The authors generated their own ASL database, this was captured in five people covering 24 letters of the alphabet, and each person's letters had about 528 photos.

**Table 20.** State-of-the-Art Comparison.

| Reference | Recognition Rate (%) | Dataset |
|---|---|---|
| Y. Zhao and L.Wang [40] | 89.32 | ASL own |
| D.Rathi [41] | 95.03 | ASL MNIST |
| L.Y.Bin y Y.Huann [42] | 95.00 | ASL own |
| R. Dionisio [39] | 97.64 | ASL MNIST |
| PSO-CNN-I | 99.98 | ASL MNIST |
| PSO-CNN-II | 99.80 | ASL MNIST |

Rathi [52] presents an optimization of the transfer learning model (based on CNN) and it was applied to the ASL MNIST database, using 27,455 images of 24 letters of the ASL alphabet. The data split was as follows, 80% of the data was for training, 10% for testing, and 10% of data for validation purposes with a training batch size of 100. The best recognition rate evidenced by the author was a value of 95.03%.

In Bin et al. [53], an architecture of four convolutional layers and two pooling layers was presented. The database was generated by the researchers themselves, taking characteristics of the ASL MINIST and consisting of 4800 images; the best accuracy reported by the authors was 95.00%.

Dionisio et al. [54], reported a recognition rate of 97.64% for the ASL MNIST with a six-layer convolutional architecture, three pooling layers, a filter size of $3 \times 3$, and a batch size of 128. The database was divided using 10% of data for phase testing, 10% for phase validation, and 80% for training.

Finally, we present the recognition rate achieved by our two approaches PSO-CNN-I and PSO-CNN-II with the best values of recognition rates of 99.98% and 99.80%, respectively. In the PSO-CNN-II, the best architecture obtained for the ASL MINIST was of two layers with 117 filters per layer with a size filter of $7 \times 7$ and batch size of 129. For the PSO-CNN-I, it was of two layers with 117 filters per layer with a size filter of $7 \times 7$ and batch size of 129.

As one can observe in Table 20, the highest performance was obtained by the proposed model (PSO-CNN-I) with a value of 99.98%, achieving an advantage over the rest of the approaches.

## 6. Conclusions and Future Work

In summary, in this paper, we present two approaches to optimize CNN architectures by implementing the PSO algorithm, these being applied to sign language recognition. The main contribution was to find some CNN hyper-parameters; in the proposals the number of convolutional layers, the size of the filter used in each convolutional layer, the number of convolutional filters, and the batch size were included. According to the experimentation and the results obtained in the two PSO-CNN optimization methodologies, we can conclude that the recognition rate increased in all case studies carried out, providing a robust performance with the minimum parameters. Overall, the recognition rates achieved by the three databases were as follows: for the ASL MNIST database, the best value was 99.98% and an average of 99.53% with the PSO-CNN-I approach. For the ASL alphabet

database, the best accuracy was 99.87% and an average of 99.58% with PSO-CNN-I, and for the MSL alphabet, the best value was 99.45% and an average of 98.91% after applying the PSO-CNN-II approach. After a comparative analysis against other state-of-the-art works focused on sign language recognition (ASL and MSL), we can confirm that the optimization approaches of this work present competitive results.

This research focused on optimizing the number of convolutional layers, the filter size used in each convolutional layer, the number of convolutional filters, and the batch size. The results provide evidence of the importance of applying optimization algorithms to find the optimal parameters of convolutional neural network architectures.

As future work, the PSO algorithm could be applied to optimize other CNN hyper-parameters, implement another version of the PSO algorithm or explore different evolutionary computational techniques, to produce more robust CNN architectures that will be implemented in different sign language datasets used in other countries. In the experimental test, the images were introduced as static images, but we are considering working with input images in real-time or capturing them through video. On the other hand, our idea is to be able to implement the use of this proposal in the development of assisted communication tools and to contribute to human−computer iteration applications that can be of support to the deaf community.

**Author Contributions:** Individual contributions by the authors are the following: formal analysis, C.I.G.; conceptualization, writing—review and editing G.E.M. and C.I.G.; methodology, G.E.M.; investigation, software, data curation, and writing—original draft preparation, J.F. All authors have read and agreed to the published version of the manuscript.

**Funding:** This research received no external funding.

**Acknowledgments:** We thank CONACYT for the financial support provided with the scholarship number: 954950 and our gratitude to the program of the Division of Graduate Studies and Research of the Tijuana Institute of Technology.

**Conflicts of Interest:** The authors declare no conflict of interest.

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
