# Peer review of "Optimization of Convolutional Neural Networks Architectures Using PSO for Sign Language Recognition"

_axioms, doi:10.3390/axioms10030139_

Round 1

Reviewer 1 Report

The authors presented a PSO algorithm for designing of CNN. The algorithm searches for  optimal values of the networks hyper parameters. They propose and analyse two variants of the objective. The optimised architectures are applied on three studies cases of sign language databases.  I have following remarks:

  1. Line 213 it is written that Equation (1) updates the velocity and Equation (2) the position. After in line 215 is written that Equation (1) is the position. I think the line 213 is wrong.
  2. In Equation (1) is used xi and pi, but in Equation (2) they are xij and pij, which confuse the reader.

Author Response

Dear editor and reviewers, Thank you for your useful comments and suggestions on our manuscript. We have modified the manuscript accordingly. We have provided a marked copy clearly indicating the revisions. In the revised manuscript, we marked the revisions for response to reviewers’ comments with red color. The detailed corrections are listed below point by point.

Reviewer #1:

The authors presented a PSO algorithm for designing of CNN. The algorithm searches for optimal values of the networks hyper parameters. They propose and analyze two variants of the objective. The optimized architectures are applied on three studies cases of sign language databases.  I have following remarks:

  1. Line 213 it is written that Equation (1) updates the velocity and Equation (2) the position. After in line 215 is written that Equation (1) is the position. I think the line 213 is wrong.
  • We thank reviewer 1 for this comment. We have solved this problem by making the correct assignment of each equation.
  1. In Equation (1) is used xi and pi, but in Equation (2) they are xij and pij, which confuse the reader.
  • We appreciate this comment. We have reviewed the literature again and we have fixed this issue by correcting Equation (2).

Reviewer 2 Report

The authors propose PSO approaches for designing convolutional neural networks. Experiments are held with three study cases on the domain of sign languages.

All in all the methodology is well described and the experimental part looks sound. I have only few minor points to report as follows.

1) The authors may introduce a proper related work section and mention PSO applications to the machine learning field such as e.g.

  • An optimisation-driven prediction method for automated diagnosis and prognosis, 2019
  • Employing artificial bee colony and particle swarm techniques for optimizing a neural network in prediction of heating and cooling loads of residential buildings, 2020
  • ...

2) In section 5.5, the authors adopt the Z-test which assumes a normal distribution on the numerical data. However, usually the experimental results of metaheuristics are not normally distributed. If the authors want to use Z-test they need to perform a normality test before (e.g. Kolmogorov-Smirnov). Another and better possibility is to perform a non-parametric test such as the Wilcoxon test. More information about statistical test can be found in this tutorial:

  • A practical tutorial on the use of nonparametric statistical tests as a methodology for comparing evolutionary and swarm intelligence algorithms, 2011

Author Response

Dear reviewers, Thank you for your useful comments and suggestions on our manuscript. We have modified the manuscript accordingly. We have provided a marked copy clearly indicating the revisions. In the revised manuscript, we marked the revisions for response to reviewers’ comments with red color. The detailed corrections are listed below point by point.

Reviewer #2:

The authors propose PSO approaches for designing convolutional neural networks. Experiments are held with three study cases on the domain of sign languages.

All in all the methodology is well described and the experimental part looks sound. I have only few minor points to report as follows.

1) The authors may introduce a proper related work section and mention PSO applications to the machine learning field such as e.g.

  • An optimisation-driven prediction method for automated diagnosis and prognosis, 2019
  • Employing artificial bee colony and particle swarm techniques for optimizing a neural network in prediction of heating and cooling loads of residential buildings, 2020.
  • In the new version of the paper, we have added some statements in the introduction to include related work, including the application of PSO in the field of machine learning. Recommended research papers were also cited as references [27] and [28]. Thanks to the reviewer for this relevant suggestion to improve our work.

2) In section 5.5, the authors adopt the Z-test which assumes a normal distribution on the numerical data. However, usually the experimental results of metaheuristics are not normally distributed. If the authors want to use Z-test they need to perform a normality test before (e.g. Kolmogorov-Smirnov). Another and better possibility is to perform a non-parametric test such as the Wilcoxon test. More information about statistical test can be found in this tutorial:

  • A practical tutorial on the use of nonparametric statistical tests as a methodology for comparing evolutionary and swarm intelligence algorithms, 2011
  • On this point, we agree with the reviewer that the results generated by metaheuristic algorithms are generally not normally distributed. Therefore, after analyzing the suggestion and reviewing the recommended tutorial (on nonparametric statistical tests), we decided to implement Wilcoxon's nonparametric test. In the new version of the article, we replace the Z-test with the Wilcoxon test. We present the analyzes in Section 5.5. (5.5. Statistical test between the PSO-CNN-I and PSO-CNN-II optimization process). Thank you for these useful suggestions.

The revised manuscript has been resubmitted to your journal. We look forward to your positive response soon. Thank you very much for your time and consideration.

Round 2

Reviewer 1 Report

All reviewers recommendations are taken in to account. The paper can be accepted in its present form.